# FR-PatchCore: An Industrial Anomaly Detection Method for Improving Generalization

**DOI:** 10.3390/s24051368

**Published:** 2024-02-20

**Authors:** Zhiqian Jiang, Yu Zhang, Yong Wang, Jinlong Li, Xiaorong Gao

**Affiliations:** School of Physical Science and Technology, Southwest Jiaotong University, Chengdu 611756, China; 2022201064@my.swjtu.edu.cn (Z.J.); wangyonga@swjtu.edu.cn (Y.W.); jinlong_lee@126.com (J.L.); gxrr@vip.163.com (X.G.)

**Keywords:** image anomaly detection, self-supervised learning, feature processing

## Abstract

In recent years, a multitude of self-supervised anomaly detection algorithms have been proposed. Among them, PatchCore has emerged as one of the state-of-the-art methods on the widely used MVTec AD benchmark due to its efficient detection capabilities and cost-saving advantages in terms of labeled data. However, we have identified that the PatchCore similarity principal approach faces significant limitations in accurately locating anomalies when there are positional relationships between similar samples, such as rotation, flipping, or misaligned pixels. In real-world industrial scenarios, it is common for samples of the same class to be found in different positions. To address this challenge comprehensively, we introduce Feature-Level Registration PatchCore (FR-PatchCore), which serves as an extension of the PatchCore method. FR-PatchCore constructs a feature matrix that is extracted into the memory bank and continually updated using the optimal negative cosine similarity loss. Extensive evaluations conducted on the MVTec AD benchmark demonstrate that FR-PatchCore achieves an impressive image-level anomaly detection AUROC score of up to 98.81%. Additionally, we propose a novel method for computing the mask threshold that enables the model to scientifically determine the optimal threshold and accurately partition anomalous masks. Our results highlight not only the high generalizability but also substantial potential for industrial anomaly detection offered by FR-PatchCore.

## 1. Introduction

The issue of data imbalance is widely observed in industrial anomaly detection, further exacerbated by the scarcity of valuable anomalous data available for this task. Anomalies can arise from various unknown external influences [1], and they can also originate from the objects themselves. It is impractical to account for every possible exception. Unsupervised approaches address this challenge by leveraging unlabeled samples to enable the model to learn the distribution properties and features of both normal and abnormal samples. This enables effective anomaly detection while avoiding the high cost associated with labelled data. In the field of industrial anomaly detection, training autoencoders [2,3] and methods based on generative adversarial networks (GANs) [4,5,6,7] are commonly employed. Lyu et al. [8] fused GAN with a deep convolutional neural network (DCNN) to build an anomaly location model, which can find the mapping relationship between images and high-dimensional features. Son et al. [9] computed anomaly scores by training encoder–decoder-based long short-term memory (LSTM) networks to evaluate data based on the time the anomaly existed. J. Mitra et al. [10] used generative adversarial networks to identify a manifold of normal samples and, at the same time, identify abnormal patterns that fall outside the model. Xiao, K., Cao, J., Zeng, Z., and Ling, W.-K. [11] explore sample distribution by utilizing the fast-labelling characteristics of a graph structure and use an autoencoder to blur low-level features while retaining local features to achieve better anomaly detection performance. However, these methods often rely on the availability of a substantial amount of data. When the number of anomalous samples is limited, the model may struggle to distinguish between normal and anomalous images accurately.

Consequently, self-supervised approaches are highly favored in the field. Several state-of-the-art self-supervised methods, such as PatchCore [12], have achieved remarkable detection results on the widely recognized MVTec AD benchmark [13]. However, it has been observed that the PatchCore method imposes stringent requirements for image alignment. It is worth noting that in the MVTec dataset, most categories satisfy the condition of “similarity”, where samples within the same category share common characteristics and can be approximately aligned at identical positions. These samples do not exhibit significant spatial differences, such as rotation or flipping. Hence, they demonstrate excellent test results and are referred to as “similar categories”. In contrast, categories that fail to meet these similarity conditions are termed “spatial transformation categories”. The effectiveness of their anomaly detection may be compromised as we have observed that these methods are not proficient in handling the latter type of category.

To establish that this is not a random occurrence, we also conducted experiments on the MPDD dataset [14], whose six categories also have “spatial location relationships”. These experiments confirmed that the PatchCore method is more sensitive to “similar pictures” and does not pay much attention to the spatial position relationship between samples. In industrial detection, the images to be analyzed are not limited to “similar categories”. They can be captured by sensors from various angles and locations. Moreover, when training data are scarce, employing data augmentation techniques to expand the dataset (e.g., rotation, mirror flipping, left, and right flipping, etc.) can introduce “spatial transformation categories”, which may reduce the effect of anomaly detection, resulting in increased rates of missed detection and false positives.

To enhance the generalization performance of the method and enable it to effectively detect even in the presence of “spatial transformation categories”, we introduce registration as a pretext task. We extract fused features from the feature extraction layer and store them in a database through a series of dimensionality reduction operations. By utilizing a negative cosine similarity loss, we update the features in the memory bank, eliminating overlapping features while incorporating previously neglected ones during training. As the tests adopt a pixel-by-pixel comparison approach, disregarding pixel relationships, we construct a module similar to pyramid pooling to enforce connectivity between patches. Additionally, we devise an innovative approach that enables the model to look for the optimal threshold, resulting in accurate ground truth segmentation. Our proposed method achieves an impressive AUROC performance of 98.81% on the MVTec dataset, significantly enhancing generalization and reducing false positive rates.

## 2. Related Works

Recently, supervised methods have exerted a significant impact on the field of anomaly detection, resulting in the assimilation of concepts from these approaches into self-supervised methods. Self-supervised anomaly detection methods can be broadly classified into reconstruction-based techniques and feature embedding-based techniques.

### 2.1. Reconstruction-Based Approaches

Reconstruction-based anomaly detection relies on the assumption that the model can reconstruct normal samples well but struggles with anomalous regions. Early attempts involved using generative adversarial networks (GANs) [15,16]. However, due to the strong generalization of neural networks, abnormal samples could also achieve good reconstruction, rendering the comparison unreliable. More recently, You et al. [17] introduced a new paradigm for reconstruction networks, addressing the issue where popular reconstruction networks may fall into a “same shortcut” scenario, where both normal and abnormal samples can be accurately recovered, leading to difficulties in detecting outliers. This work tackles the practical challenge of detecting anomalies across different object classes using a unified framework.

### 2.2. Feature Embedding-Based Approaches

The feature embedding-based approach involves feeding images into the model, extracting features, and constructing interface scoring rules in the feature space. Unlike the reconstruction method that operates in the RGB image space, feature embedding focuses on the high-dimensional feature space for anomaly detection. Notably, Bergman, L., Cohen, N. and Hoshen, Y. [18] encouraged the use of pre-trained models on large-scale datasets for anomaly detection. This recommendation gives great encouragement to such methods.

SPADE [19] initially extracts features of all normal samples (training set) in a network pre-trained on ImageNet. During testing, the K-nearest neighbors (KNN) [20] metric is applied to each pixel-wise feature of the test sample to calculate the anomaly score for each pixel point. However, the more normal images used in training, the more features are stored, resulting in higher KNN complexity during testing. Building upon SPADE, a method named PaDiM [21] improves it by eliminating the construction of a feature bank and performing KNN for anomaly detection. However, the pixels at each position are not strictly aligned, so modelling each pixel position alone may lead to inaccuracies. Y. Zheng et al. [22] performs fine-tuning based on the anomaly localization task and proposed FYD. To address the feature misalignment issue in PaDiM, they introduce the spatial transformer network (STN) [23] module for coarse alignment and use a self-supervised learning paradigm inspired by the SimSiam network (without requiring negative samples). However, certain images may still not be aligned. Furthermore, from a visualization perspective, the detection performance is not optimal, primarily due to the evaluation metric (pixel-wise AUCROC score [24]), which tolerates anomalies in small areas.

Recently, Huang et al. [25] innovated based on PaDiM and FYD and proposed RegAD, which employs registration as a proxy task to train an anomaly detection model with unknown categories. PatchCore addresses the slow testing speed of SPADE. KNN and greedy coreset subsampling are applied to select the most representative feature points for measuring anomaly scores during testing, which reduce the size of the feature bank and finally achieve excellent results.

The method FR-PatchCore (Feature-level Registration PatchCore) of ours preserves the high efficiency of PatchCore. Feature-level registration is employed as a pretext task, and the information stored in the feature database is updated using a negative cosine similarity loss. To enforce pixel relationships, a module similar to pyramid pooling is constructed. During testing, the Euclidean distance from each pixel to its corresponding feature in the database is used for scoring.

## 3. Methodology

In our methodology, we introduce the concept of feature-level registration and utilize it as a non-cumbersome pretext task during training. The negative cosine similarity loss is employed as the training loss, which we optimize to update the memory bank. The framework of our approach is depicted in Figure 1.

The upper section represents the registration module, while the lower section represents the memory bank module. Both sections operate concurrently. Two images from samples belonging to the same category are randomly selected, and their features are extracted using the convolutional neural network and spatial transformer network (CNN + STN) for registration purposes. The feature registration process is supervised by maximizing the absolute value of the negative cosine similarity loss. After the registration process, the extracted fusion features are stored in a dedicated memory bank and continuously updated with training losses after each registration iteration.

### 3.1. Registration Module

Neural network training necessitates task-driven learning. Therefore, the essence of self-supervised learning lies in the thoughtful design of tasks that facilitate effective model learning. Inspired by the work of [25], which obtained a Gaussian distribution model of normal data through feature-level registration training, we leverage the registration task as a pretext task to enhance the model’s understanding of features and emphasize spatial and positional differences. Accordingly, we construct the registration module, consisting of a feature extractor, feature encoder, and predictor, as illustrated in Figure 2.

In feature registration, since spatial transformation can be represented as matrix operations, it is advantageous to allow the network to learn the generation of matrix parameters, thereby acquiring spatial transformation capabilities. Common network frameworks in deep learning include the CNN and the transformer. Additionally, the spatial transformer network, which plays a pivotal role in our research, can seamlessly integrate into any component of the CNN architecture. To ensure a fair comparison with state-of-the-art methods, we have selected the wide_resnet_50_2 [26] network as the backbone for our experiments among various CNNs commonly employed in anomaly detection tasks. This network has demonstrated exceptional performance on the ImageNet dataset, achieving an accuracy of 78.51% for Top 1 and 94.09% for Top 5. For this specific task addressed in our paper, we conducted an ablation experiment (4.5.1) to compare the feature extraction capabilities of both Resnet and VIT models, where wide_resnet_50_2 exhibited superior performance. 

We incorporated the *STN* module into the wide_resnet_50_2 architecture. The overall structure of the *STN* module is illustrated in Figure 2 and consists of a localization network, a grid generator, and a sampler. In the first component, a feature map is given as Formula (1):(1)$$U\in R^{H\times W\times C}$$

After several convolutions or fully connected layers, a regression layer follows, leading to the output of regression transformation parameter θ. The dimension of θ depends on the specific transformation type chosen by the network.

In the second component, we utilize θ and the specified transformation mode from the localization network output, and the grid generator performs further spatial transformations of the feature map to determine the mapping of ***T(θ)*** between the output and input features. It employs the predicted transformation parameters to create a sampling grid, which represents a set of points where the input map should be sampled to produce the transformed output. In the sampler, it utilizes the sampling grid to determine which points in the input feature map will be utilized for the transformation. It samples the input feature map against the sampling grid to obtain the final output. 

In Step 1, assuming that the input RGB image has a resolution of (224,224), Formula (2) is applied.
(2)$$U\in R^{224\times 224\times 3}$$

The fourth layer of the wide_resnet_50_2 network is excluded to preserve more comprehensive spatial information, and the spatial transformer network (STN) is integrated after the initial three layers of the network. The input feature $f_{i}(x_{i}^{s},y_{i}^{s}$) undergoes a transformation function $\tau_{\theta }$. The mapping relationship between the input and output feature mappings is defined as Formula (3).
(3)xisyis=τθGi=Aθxityit1=θ11θ12θ13θ21θ22θ23xityit1

In this case, the form of the eigenvector is shown in Formula (4):(4)$$\left\{\begin{array}{c} U_{1}\in R^{56\times 56\times 64}\\ U_{2}\in R^{28\times 28\times 128}\\ U_{3}\in R^{14\times 14\times 256} \end{array}\right.$$

When no key points are labeled, the STN allows the neural network to actively transform the feature map based on input features and learn spatial transformation parameters without requiring additional training supervision or modifications to the optimization process. As illustrated in Figure 2, the STN can effectively align input images or learned features during training, thereby mitigating the impact of spatial geometric transformations such as rotation, translation, scale, and distortion on tasks like classification and localization. The STN facilitates the spatial transformation of the input data, thereby enhancing feature classification and enabling the network to achieve rotational invariance dynamically. Moreover, it intelligently selects the most salient region of the image and optimally transforms it into a suitable orientation. Figure 2 illustrates the depiction of feeding an inverted screw image into the STN module. Through a series of transformations, the input is effectively rectified to face forward.

The employed approach in Figure 2 involves the utilization of a Siamese network for feature encoding, wherein a negative cosine similarity loss is applied as per Formulas (5) and (6).
(5)$$D\left(p_{a},z_{b}\right)=-\frac{p_{a}}{{\left\|p_{a}\right\|}_{2}}\cdot \frac{z_{b}}{{\left\|z_{b}\right\|}_{2}}$$
(6)$$D\left(p_{b},z_{a}\right)=-\frac{p_{b}}{{\left\|p_{b}\right\|}_{2}}\cdot \frac{z_{a}}{{\left\|z_{a}\right\|}_{2}}$$

The negative cosine similarity loss is an appropriate metric for quantifying the similarity between two vectors. Furthermore, it possesses the capability to map similar vectors to adjacent points and dissimilar vectors to distant points. This characteristic facilitates feature clustering, as described in Section 3.2.3.

The objective is to maximize the similarity between $p_{a}$ and $z_{b}$, as well as between $p_{b}$ and $z_{a}$. To prevent the input data from converging to a constant value after convolution activation, resulting in identical outputs regardless of the input image, we adopt the approach described in [27] by halting the gradient operation on one of the branches to avoid model collapse. Finally, we define Formula (7) as the registration loss for symmetric features.
(7)$$L=\frac{1}{2}[(D\left(p_{a},z_{b}\right)+(D\left(p_{b},z_{a}\right)]$$

The STN is employed in this stage to perform feature rotation and inversion, facilitating the model’s determination of image similarity. Following each training iteration, a negative cosine similarity loss is obtained.

### 3.2. Memory Bank Acquisition

#### 3.2.1. Feature Extraction

Let φ(h,w,c) represent the feature mapping of the second layer, where h denotes the height, w denotes the channel width, and c denotes the number of channels. Like PatchCore, the patch-level features of local features in clustered neighborhoods can be represented as Formula (8):(8)$$Ρ=f_{agg}\left(\varphi \right)$$

Here, $f_{agg}$ represents the aggregation function within the neighborhood. As mentioned in Figure 1, we use a combination of the first three layers of wide_resnet_50_2 and *STN* to extract features and build the memory library, with each layer followed by the *STN*. Inspired by the PatchCore method, we likewise did not adopt the last layer of the Resnet network since it lost a lot of the features’ spatial information. As shown in Figure 3, the features of the second and third layers of the *STN* can retain global information while containing more local feature information. However, if the features of the first three or more layers are fused, the features in the input memory library will not contain enough information for accurate detection. 

#### 3.2.2. Similar to Pyramid Pooling Module (SPPM)

After feature extraction, a 3-dimensional tensor with the shape (C,H,W) is obtained, where C is the sum of the dimensions of the second and third layers. We then process this three-dimensional feature vector by flattening it in three dimensions except for the channels. This results in a two-dimensional tensor with a shape of (H×W,C), which is randomly projected. Thus, we flatten the original feature vector into a column-wise phenotypic feature matrix. However, as mentioned earlier, the PatchCore method employs a pixel-by-pixel search approach, conducting nearest neighbor searches on each pixel and disregarding pixel relationships. We recognize that 2D average pooling can increase the receptive field, which is crucial for anomaly detection tasks. Hence, we employ the pooling approach illustrated in Figure 4.

Assuming an input feature size of 64 × 64, we perform three pooling operations with pool core sizes of 3 × 3, 4 × 4, and 5 × 5. Subsequently, tensors of sizes 64 × 64, 32 × 32, and 16 × 16 are obtained. Next, the three pooled tensors are unsampled to the same size of 64 × 64, and their dimensions are concatenated. The earlier pooling regions are included within subsequent pooling regions using this approach to enlarge the receptive field and give more attention to edge information due to an overlap between pooling regions, thus establishing closer relationships between the pixels. 

After the pooling step, the eigenvector of the domain φ(h,w), as represented by Formula (9), is enacted.
(9)$$N_{p}^{\left(h,w\right)}=\left\{\left(a,b\right)|a\in \left[h-\frac{p}{2},\ldots h+\frac{p}{2}\right],b\in \left[w-\frac{p}{2},\ldots w+\frac{p}{2}\right]\right\}$$

The feature aggregation operation is utilized to obtain the locally aware patch feature set p of the feature mapping tensor $\varphi_{i,j}$, enabling the successful realization of the clustering ${Ρ}_{s,p}\left(\varphi (h,w)\right)$ of the feature tensor, as shown in Formula (10).
(10)$${Ρ}_{s,p}\left(\varphi (h,w)\right)=\left\{\varphi (h,w)\left({{Ν}_{p}}^{\left(h,w\right)}\right)\right|h,\;w\;mod\;s=0,h<h^{*},w<w^{*},h,w\in Ν\}$$

In this case, the feature repository $M_{1}$ can be described as Formula (11):(11)$$M_{1}=\underset{xi\in N}{⋃}{Ρ}_{s,p}\left(\varphi (xi\right)$$

#### 3.2.3. Anomaly Detection

The samples used in self-supervised learning are exclusively normal. The training process aims to identify representative features of the “normal category” and utilize them as a reference for evaluating positivity and negativity, ultimately achieving anomaly detection. During testing, we index the memory bank that stores characteristic information of positive samples and calculate the Euclidean distance between sample patches to obtain an anomaly score. In n-dimensional space, if there are two points $a(x_{11},x_{12},\ldots ,x_{1n})$ and $b(x_{21},x_{22},\ldots ,x_{2n})$, then the Euclidean distance d is defined as Formula (12).
(12)$$d=\sqrt{\sum_{k=1}^{n}{{(x}_{1k}-x_{2k})}^{2}}$$

After completing the feature aggregation and pooling module in Step 2, we proceed to the feature clustering operation. The utilization of negative cosine similarity loss during training greatly facilitates feature clustering by ensuring that similar vectors are consistently mapped to proximate locations with each iteration, while dissimilar vectors are assigned to distant points. This approach effectively reduces the clustering time and enhances the operational efficiency of the model. To optimize the memory library size and improve the testing efficiency, we implement the PatchCore method and employ greedy subsampling for reducing and optimizing the memory bank based on Formula (13).
(13)$$M_{2}^{*}=\frac{\arg min}{M_{2}\subset M_{1}}\frac{max}{m\subset M1}\frac{min}{n\in M_{2}}{\left\|m-n\right\|}_{2}$$

The objective of our purpose is to streamline the testing process by solely conducting nearest neighbor searches for the test sample’s feature in $M_{2}$, identifying its closest neighboring feature and subsequently calculating the maximum Euclidean distance from this feature to its clustering center. This approach allows us to obtain an anomaly score, facilitating effective anomaly detection.

In the memory bank, for each patch-level feature $m\in M_{2}$ of the training data, we select the $m^{*}$ nearest neighbors from the patch-level features of the test data. The patch-level anomaly score of the test image $X_{test}$ is estimated based on the distance between the patch-level feature $m_{test}$ and $m^{*}$, as shown in Formulas (14) and (15).
(14)$$m^{*}={arg\;min}_{m\in M_{2}}\;\;\;{\left\|m^{test}-m\right\|}_{2}$$
(15)$$s^{*}={\left\|m^{test}-m^{*}\right\|}_{2}$$

#### 3.2.4. Post-Processing Method

In anomaly detection, the ground truth serves as a baseline measurement obtained from a reliable method. It is used to calibrate and improve the accuracy of new measurement methods. Many current anomaly detection methods rely on the calculated AUROC curve to determine the threshold of the entire dataset. This involves setting the threshold based on the data distribution and other characteristics. The regions above the threshold are considered abnormal, while those below are considered normal. However, if the overall threshold of an abnormal image is lower than the dataset threshold, the anomaly cannot be detected. Otherwise, false detections may occur. To address this, we propose a threshold determination method based on the image itself, which identifies the critical point between normal and anomalous regions, enabling the accurate labelling and localization of anomalies.

For each test sample, as shown in Figure 5a, we artificially construct five rectangular sampling frames in its upper left, lower left, upper right, lower right, and right center regions, as shown in Figure 5b.

The constructed image is also fed into the network for feature extraction, and then we search for distances in the feature database, compute scores, and obtain the score matrix. Its heat map visualization is shown in Figure 5c. Obviously, as artificially constructed anomalies, these five areas will have significantly higher anomaly scores than the rest of the picture. We set the pixel values of these five areas to 0 and the other parts to 1, thus obtaining its ground truth, as depicted in Figure 5d. Then, we traverse the five regions, find the maximum and minimum fractional values (${thres}_{max}\;\mathrm{and}\;{thres}_{min}$) and define the ***interval*** as outlined in Formula (16), setting ***iter_times*** to 10 to ensure the code runs efficiently:(16)$$interval={(thres}_{max}{-thres}_{min})/iter\_times$$

For each iteration, we set the threshold as outlined in Formula (17):(17)$$th={thres}_{max}{-iter\_times\cdot thres}_{min}$$

The iteration is carried out in the range of minimum and maximum thresholds, and each th can output an anomaly mask accordingly, where the white area is abnormal and the black area is normal, as shown in Figure 6a–e, which are the masks corresponding to the thresholds 9.0, 8.3, 7.6, 6.4 and 6.1, respectively. As can be seen from the five pictures shown in Figure 6, with the continuous increase in the threshold, the area of the abnormal region in the mask image corresponding to each threshold value increases first and then decreases. Finally, we calculate the intersection over union (IOU) between the prediction box and the ground truth box. The threshold is updated according to the highest IOU, and the best threshold (***best_th***) is obtained after 10 iterations. The mask graphs corresponding to different responsiveness are presented in Figure 6a–e, serving the purpose of facilitating the description of this method’s principle. It is important to note that these graphs do not represent the final segmentation results but rather serve as temporary variables within the code and are subsequently deleted after calculation to optimize memory usage.

After determining the optimal threshold value (***best_th***), based on industrial detection expertise, we establish the region of abnormal pixels (S = 25) and devise an algorithm. For the neighboring pixel region exceeding ***best_th***, if it is smaller than S, the pixel value in this region will be set to 0 (normal); otherwise, it will be set to 1 (abnormal). The implementation of this setting will effectively eliminate the occurrence of false positives in small areas, resulting in a reduction in the overall false alarm rate observed in the test results.

## 4. Experiment

### 4.1. Datasets and Metrics

We first evaluate the performance of our method on two public datasets, MVTec and MPDD, which cover both classical industrial anomaly detection and defect detection in manufacturing processes.

-MVTec dataset: The MVTec dataset is a well-known benchmark for industrial anomaly detection. It consists of 15 categories with a total of 3629 training and validation images and 1725 test images. The training set contains only normal images, while the test set includes images with various defects and normal images. The images have resolutions ranging from 700 × 700 to 1024 × 1024 pixels, and pixel-level ground truth labels are provided for each defect.-MPDD dataset: The MPDD dataset is a newly proposed dataset focused on defect detection in the manufacturing process of painted metal parts. It comprises six types of metal parts captured under different spatial orientations, positions, distances, light intensities, and backgrounds. We utilize this dataset to evaluate the registration effect of our method and compare it with several other methods to highlight the superiority of our approach.-Industrial scenario dataset: To demonstrate the robustness and generalization of our approach, we have selected an additional three datasets from the field of high-speed rail (HSR) component inspection for experimentation. Each of these industrial datasets consists of 460 normal samples and 115 abnormal samples, all of which exhibit visual bias in terms of their positioning. Figure 7 showcases the appearance of these normal samples.

We first adopt the traditional model evaluation method, randomly sampling normal data and abnormal data, and divide the dataset according to the ratio of training set to test set of 8:2. In order to avoid overfitting and other effects caused by unreasonable dataset division, we adopt the cross-validation method to evaluate the model in Section 4.4.

For the evaluation, we employ two threshold-independent metrics. The first is the area under the receiver operating characteristic curve (AUROC), which measures the true positive rate (percentage of correctly classified anomalous pixels). The image-level AUROC scores assess the model’s detection accuracy, denoted as detection (Det.), while the pixel-level AUROC scores evaluate the model’s positioning and segmentation accuracy, indicated as segmentation (Seg.). The second metric is the per-region-overlap score (PRO-Score) [28], which considers the mean rate of correctly classified pixels as a function of the false positive rate between 0 and 0.3 for each connected component, as shown in Formula (18). Higher scores indicate better localization of both large and small anomalies [29].
(18)$$PRO=\frac{1}{N}\sum_{n=1}^{N}\frac{P\cap G_{n}}{G_{n}}=\frac{1}{N}\sum_{n}^{N}\frac{{TP}_{n}}{{TP}_{n}+FN_{n}}$$
where P represents the prediction result, $G_{n}$ represents the true value, then the intersection of the two is the true positive sample ${TP}_{n}$, and $FN_{n}$ represents the false negative sample.

The above two indicators are commonly employed in anomaly detection tasks and are widely acknowledged and utilized as benchmarks within the detection industry. However, due to the intricacy of industrial detection scenarios, there exists variation in data quality as well as resolution. To demonstrate the robustness of our method, we also employ three additional indicators, namely the recall rate, accuracy rate, and false alarm rate (***FAR***), for testing and cross-validation using the high-speed railway dataset. The calculation formulas for these indicators are presented in Formulas (19)–(21), respectively.
(19)$$Recall=\frac{TP}{TP+FN}$$
(20)$$Accuracy=\frac{TP+TN}{TP+TN+FP+FN}$$
(21)$$FAR=\frac{FP}{FP+TN}$$

The term ***FP*** denotes the number of false positives, while ***TN*** represents the count of true negatives.

### 4.2. Implementation Details

We utilize wide_resnet_50_2 and the *STN* as the underlying architecture to construct a convolution-based encoder and predictor. These models are trained on images with a resolution of 224 × 224 using an Asus NVIDIA GTX 3090 GPU sourced from Chengdu, China. For parameter updates, we employ momentum SGD for 55 epochs with a learning rate of 0.0001 and a batch size of 32. The duration of each training round is approximately 8 min, with an average training epoch lasting for about 8.7 s.

### 4.3. Anomaly Detection

To assess the efficacy of FR-PatchCore, we conduct a comparative analysis with other cutting-edge anomaly detection techniques, such as SPADE, PaDiM, Patch-SVDD [30], Cutpaste [31], RegAD, and PatchCore. We meticulously replicate their reported outcomes using each method’s official source code to ensure an equitable comparison. The last row displays the macro-averaged scores for all categories, highlighting the highest scores in bold.

The MVTec dataset is the most widely used benchmark for anomaly detection, and FR-PatchCore achieves a 98.81% image-level AUROC score and a 97.86% pixel-wise AUROC score when compared with the other six methods in Table 1. In the comparison of detection results of “similar categories” such as bottle, leather, cable, etc., our method is slightly inferior to PatchCore, but it still has high detection scores and accuracy, and the average detection scores of all categories are better than the other six methods. For the three “spatial transformation classes” such as hazelnut, screw, and metal nut in the dataset, we present partial representative results of their visualizations in Section 4.6.

To evaluate the robustness of our method against the “rotation class” data that we defined, we conducted experiments on the MPDD dataset, which are displayed in Table 2, from ten runs. The final row presents the macro-averaged scores across all categories, with the best scores indicated in bold.

We present partial representative results of the MPDD dataset visualizations in Section 4.5. Unlike the MVTec dataset, all six categories in the MPDD dataset have spatial differences and position shifts. This is challenging for most methods because anomaly detection models are not trained enough to deal with morphologically diverse samples. We believe that increasing the training time is not enough to make the model learn more. On the contrary, it may cause the model to overfit. Most models cannot detect the “spatial transformation class” of the sample effectually. The FR-PatchCore proposed by us is innovative in model learning. As shown in Table 2, our method leads in comparison with the other six methods. This proves that the idea of using feature-level registration as a pretext task is worthwhile. Only when the model learns the spatial information that has been ignored before can it better deal with such problems, improve the generalization and be applied to the field of high-standard industrial anomaly detection.

To evaluate the applicability of our method in real-world scenarios, we conducted experiments using data from high-speed rail (HSR) components in industrial anomaly detection. The scores are displayed from ten runs in Table 3.

We present partial representative results of the HSR dataset visualizations in Section 4.5. As can be seen from Table 3, FR-PatchCore still achieves an excellent detection performance even when detecting real industrial datasets with a complex environment, noisy images, and many influencing factors, which proves that our method has good generalization and robustness.

Most anomaly detection researchers evaluate the performance of their methods using both image-level and pixel-wise AUROC curves. AUROC is a widely used metric for assessing the ability of models to discriminate between abnormal and normal samples. However, in anomaly detection, it is not only important to identify anomalies but also to accurately localize them. The pixel-wise AUROC evaluation provides a broad assessment, where the score is significantly improved if a large region is correctly localized but has a minimal impact if a small region is incorrectly positioned. 

To address this limitation, we also employed the PRO score [24] criteria, which considers the rate of correctly classified pixels as a function of the false positive rate (fpr). We calculated the average PRO score of the three datasets using an fpr of 30%, following the methodology described in [25]. In Table 4, we compare our method with SPADE, PaDiM, and PatchCore, which are all based on feature embedding. Our approach achieves the highest PRO-Score, indicating a superior localization performance.

In the comparison of the PRO score, we only compare four methods based on feature embedding: SPADE, PaDiM, PatchCore and our proposed method. Since these four methods have the same basic principle, they are easier to compare, and it is more convenient to use the principle of control variables. For example, the same backbone wide_resnet_50_2 is used at the same time. As shown in Figure 8, we report the results of the PRO score comparison on three datasets of different difficulty, which indicates that FR-PatchCore has an excellent detection performance and still obtains good indicators under this more stringent score limitation.

In addition, as presented in Table 5, Table 6 and Table 7, we have documented the scores for the accuracy rate, recall rate, and false positive rate of the three high-speed rail datasets. Among the four comparative methods employed, our approach stands out as the most advanced, with the lowest occurrence of false alarms. Furthermore, apart from these metrics, industrial tests also assess efficient reasoning performance. By employing identical parameter settings across all four methods, we conducted comprehensive evaluations. The inference time considered here encompasses both inputting samples to be detected into the network for forward propagation and the processing time. The original image resolution was adjusted from 1280 × 1280 to a 224 × 224 input network size. Our method’s inference speed remains comparable to that of well-known PatchCore networks while maintaining superior anomaly detection and segmentation performance at the image-level.

### 4.4. Cross-Validation Experiment

To assess the performance of our model on limited data samples, we conducted five-fold cross-validation using the high-speed rail component dataset. The training set was divided into five folds, with four folds used for training and the remaining one for testing. Additionally, within the four-fold training data, an additional 5% of the data was extracted as a verification set. Each training and test iteration involved 442 training data, 18 validation data, and 115 test data. This approach ensures that all samples in the training set are utilized for both training and testing purposes.

Since this study is based on self-supervised learning and utilizes unlabeled positive samples for training, there is no actual labeled data available. However, it should be noted that anomaly detection inherently involves a classification task where the “normal” label is carried by the data itself during training. Therefore, in each of the five cross-validations performed, classification accuracy serves as our evaluation metric. Subsequently, we select the model with highest classification accuracy on the verification set to evaluate its performance on separate test sets through five experiments. The resulting test errors from each experiment can be obtained using Formula (22).
(22)$$E_{T}\left(\omega \right)=\sum_{x\in D}^{n}{\left(\hat{f}\left(x|\omega \right)-f\left(x\right)\right)}^{2}$$

Set ***D*** consists of N samples, where each sample $\left(x_{i},y_{i}=f\left(x_{i}\right)\right)$ represents a subset of the total set χ.

The generalization error in this case can be approximated as the average of the five test errors, as demonstrated in Formula (23).
(23)$$E_{G}\left(\omega \right)=\sum_{x\in \chi }p(x){\left(\hat{f}\left(x|\omega \right)-f\left(x\right)\right)}^{2}\approx E_{T}\left(\omega \right)$$
where p(x) represents the probability that x occurs in the total set χ.

The results of the five cross-validation iterations are presented in Table 8, Table 9 and Table 10. Taking high-speed rail parts dataset 1 as an example, the recall rate is utilized as an evaluation metric to compare the four methods in the classification task of positive and negative samples, as depicted in Table 10. The model achieving the highest score during iteration will be selected for testing purposes.

After selecting the optimal model for each of the four comparison methods, we assessed its performance by conducting tests on five separate datasets and calculated the average accuracy based on the results obtained from these five tests, as presented in Table 9. 

Finally, we computed the average test error after five iterations of the four models utilized in the testing process. It can be observed from Table 10 that our method demonstrates superior generalization capability for the HSR dataset, as evidenced by its significantly lower generalization error compared to other methods employed across the three industrial datasets.

### 4.5. Ablation Experiment

Experiments were conducted to assess the contribution of different components in our proposed approach. The MVTec dataset was used as the subject of these experiments, where we evaluated the impact of pooling, registration, and post-processing methods individually. The experimental results clearly demonstrated that our method outperformed the other methods in terms of performance.

#### 4.5.1. Backbone Selection Process

The CNN and transformer have emerged as the predominant backbone networks for deep learning in recent years, owing to their robust feature extraction capabilities. Initially designed for natural language processing, the transformer has also been adapted by researchers for image processing through vision transformer (VIT) networks. In this section, we randomly select three subgraphs to visualize the initial three layers of features extracted from VIT and wide_resnet_50_2 networks, using the screw class as an example (Figure 9 and Figure 10). Notably, the features extracted by VIT exhibit enhanced recognizability, with reduced noise compared to those obtained by wide_resnet_50_2.

The disparity in the experimental outcomes arises from variations in the scale of the training data. In contrast to VIT’s remarkable efficacy on extensive datasets, CNNs exhibit commendable feature extraction capabilities even with limited data, rendering them more amenable for deployment within data-scarce industrial domains.

#### 4.5.2. SPPM 

In the registration module depicted in Figure 1, we incorporate a 2D average pooling module alongside the CNN + STN for extracting image features. This addition serves to mitigate the over-sensitivity of the convolutional layers to positional information. We have opted not to use the SPPM in this module, as our experiments have shown that 2D average pooling yields better results. As shown in Table 11, the application of AvgPool2d in the registration module received the highest score because this module plays a crucial role in maximizing the absolute value of the negative cosine similarity loss, thereby significantly enhancing the performance of anomaly detection.

For evaluation purposes, we employ a pixel-wise anomaly detection AUROC score and PRO score as our chosen metrics. To illustrate the convergence of loss, we utilize the screw category as an example, which is depicted in Figure 11.

The convergence of the negative cosine similarity loss for the screw class is analyzed under different settings. Figure 11a demonstrates the absence of pooling results in a loss value of approximately −0.49. When our SPPM is utilized, the loss slightly increases to around −0.57, as depicted in Figure 11b. However, by incorporating 2D average pooling, the loss significantly improves and reaches approximately −0.75, as shown in Figure 11c. This observation suggests that integrating 2D average pooling enhances the model’s capacity to learn features during the registration process.

#### 4.5.3. Registration Module 

For the SPPM and the registration module, we conducted ablation experiments to assess the impact of specific components in our model, namely the SPPM and the registration module. By removing and combining these components individually, we aim to understand their contributions to the overall performance. The results in Table 12 show that when neither the SPPM nor the registration module is used, the two metrics used for evaluation have the lowest score and the lowest anomaly detection result. When both are included in the model, we will get the best anomaly detection result.

#### 4.5.4. Post-Processing Method

In Figure 12, we demonstrate the application of our proposed post-processing method to generate a visual representation of the mask. Specifically, we present an abnormal image from the high-speed rail dataset and show the original image, segmentation image, heat map, ground truth value, the mask generated without the post-processing method, and the mask generated after the post-processing method from left to right. It is evident from Figure 12e that there is a large false positive area; however, this issue is greatly reduced in Figure 12f due to our post-processing method.

The proposed post-processing method for segmentation exceptions employs intersection over union (IoU) as a metric to evaluate the test image, as depicted in Formula (24):(24)$$IoU=\frac{p_{ii}}{p_{ij}+p_{ji}-p_{ii}}$$

The true value is represented by ***i***, the predicted value is represented by ***j***, and $p_{ij}$ represents the number of pixels predicting ***i*** as ***j***. Therefore, Formula (25) is equivalent to Formula (11).
(25)$$IoU=\frac{P\cap G}{P\cup G}$$

The letter ***P*** denotes the prediction, while ***G*** represents the ground truth.

The intersection over union (***IoU***) algorithm is commonly utilized in deep learning for object detection and semantic segmentation tasks to compute the overlap ratio between different images. In Table 13, we present the average ***IoU*** metrics for the three datasets used in our experiment, while Section 4.5 showcases visualizations of selected test results. The mask graph generated after applying our proposed post-processing method exhibits a higher degree of proximity to the ground truth value, thereby resulting in an elevated average IOU value. This substantiates the efficacy of our proposed approach.

As can be seen from Figure 12e,f, the mask result obtained after segmentation via post-processing method is closer to the heat map. The responsiveness of the sample in Figure 12a is depicted in a histogram format, as illustrated in Figure 13b, while the responsiveness of the corresponding class’s normal sample is presented in Figure 13a. 

The test image predominantly consists of normal areas with a high frequency of response but a low level of responsiveness, whereas abnormal areas exhibit a low frequency of response but a high level of responsiveness. The responsivity distribution in the normal region typically conforms to an approximate normal distribution, while the responsivity in the abnormal region lies outside this distribution. The process of calculating responsiveness is analogous to binary classification. Initially, the positive and negative attributes of the image are evaluated. Subsequently, threshold-based truth rules may be employed to exclude certain normal regions exhibiting higher responsiveness and identify regions demonstrating elevated levels of responsiveness. The response value of the normal region is observed to be approximately distributed within the range of (0, 0.7), while our post-processing method calculates an optimal threshold of 0.75. The proposed approach ensures that the highly responsive normal region, depicted in Figure 13b, with a responsivity range of 0.4–0.6 is not erroneously classified as an exception, and the mask prediction yields superior results. By employing this post-processing method, the occurrence of false positives is effectively minimized.

### 4.6. Visualization

A visualization of selected categories with “spatial transformation” characteristics from various datasets used in the experiment was employed to provide a more intuitive analysis of the detection capability of FR-PatchCore, as depicted in Figure 14.

The “spatial rotation” category of MVTec and MPDD in the two public datasets is selected for display in Figure 14, lines A–F. In Figure 14b, highly accurate anomaly segmentation results are showcased, while Figure 15c visualizes the anomaly score. Once the model successfully locates and evaluates anomalies, the threshold selection method proposed by us can be utilized to obtain mask prediction results, as depicted in Figure 14e.

The anomaly detection results for three different categories in HSR 1–3 datasets are shown in line G–I of Figure 14. In contrast to the single simple scenes of the public dataset, the data captured in industrial scenes exhibit higher levels of noise and complexity in the background, making anomaly detection susceptible to environmental factors. To capture more comprehensive visual information of the inspected objects and prevent data overexposure caused by lighting or other factors during raw data collection for industrial inspection, an unfixed shooting point is adopted instead of continuous shooting from a fixed position. This leads to a significant presence of “spatial transformation” category data in industrial inspections, which further validates the necessity of our research. Based on the segmentation and prediction results from these three columns, FR-PatchCore demonstrates accurate localization of anomalies amidst complex backgrounds while exhibiting high sensitivity toward abnormal areas and low response toward normal areas. Moreover, it maintains a low false positive rate when ensuring reliable anomaly detection, thereby showcasing its potential applications and strong generalization capabilities.

### 4.7. Discussion

The AUROC and PRO scores were utilized in Table 1, Table 2, Table 3 and Table 4 to perform category-by-category anomaly detection on the three datasets employed. In comparison with other notable methods, FR-PatchCore demonstrated a superior average performance by not only achieving excellent anomaly detection for conventional “similar class” images but also effectively detecting the challenging “spatial transformation class” in the dataset.

An evaluation of three challenging datasets for high-speed rail parts, which are critical in industrial testing, was conducted using recall, accuracy, and false positive rates in Table 5, Table 6 and Table 7. Additionally, the test speed of the comparison method was assessed, and the results indicate that at an image resolution of 224 × 224, the test speed is comparable to that of the PatchCore method, with a maximum speed of 0.21 s per frame. These findings demonstrate that our approach ensures a superior recall rate and accuracy while maintaining a minimal false positive rate without compromising efficiency. 

Furthermore, to provide a more comprehensive discussion on FR-PatchCore’s capabilities, we designed a five-fold cross-validation experiment utilizing the three industrial datasets. The average error from five tests was adopted as the final evaluation index. Our proposed method exhibited the lowest average error among all tests conducted, further highlighting its competitiveness and strong generalization abilities.

In the ablation experiment conducted in Section 4.5, we performed comprehensive analyses and experiments on the utilized model modules and network selection process. Furthermore, IoU was employed as a metric to assess the effectiveness of the proposed post-processing approach. Additionally, a subset of experimental results was visually presented in Section 4.6 to illustrate the recovery process of FR-PatchCore segmentation and anomaly positioning.

The effectiveness of self-supervised learning heavily relies on a well-designed auxiliary task. To enhance the model’s understanding of local feature relationships during registration, we propose a feature-level registration task that captures and stores the maximum distance between normal features and their respective clustering centers in the feature database. In testing, precise image alignment is no longer necessary. Instead, we can simply query the memory database to calculate the maximum distance between target features and their most similar clustering centers. As a result, our approach achieves robust anomaly detection in two-dimensional images. However, it is important to acknowledge that our method also has certain limitations and disadvantages. 

In the introduction, we define the “similarity category” and “spatial transformation category” based on the spatial position relations of objects in the picture, such as rotation and flip. However, this definition is limited to two-dimensional planes and does not account for variations in depth between images that may exist across different planes. Our method is therefore not suitable for capturing spatial position relationships with significant depth differences. As illustrated in Figure 15, we present detection results of data exhibiting substantial disparities in depth information within industrial detection scenarios, along with the corresponding registration loss curve for this class shown in Figure 16. The high occurrence of false positives can be attributed to our feature-grade registration method’s inability to achieve accurate depth alignment. Figure 16 depicts the registration loss curve specifically for this category, revealing convergence when the loss reaches approximately 0.4. This indicates that our model struggles to effectively register features associated with this type of data, resulting in a scarcity of learned features compared to previous categories.

The integration of two-dimensional images with depth maps in the above scenarios can potentially yield significant benefits, as it encompasses more comprehensive spatial information. By combining these two types of images, the model can acquire an enhanced feature representation, thereby enhancing its performance. 

In addition, the proposed post-processing method is based on a single test image and determines the threshold by calculating the responsivity of each part of the image, thereby mitigating the occurrence of false positives to some extent. However, when dealing with large-scale test data and high-resolution images, this approach may introduce significant computational overhead and pose challenges for practical implementation. Furthermore, our exploration into adaptive threshold calculations remains limited, without incorporating a comprehensive and universally applicable algorithm. The responsivity values for each area of the image in Section 4.5.4 were visually represented, offering a novel perspective. If we consider the data distribution, assuming that responsiveness follows a normal distribution, then according to the three-sigma rule, an outlier is defined as a value in a set of result values that deviates from the mean value by more than three standard deviations. However, if the responsiveness does not follow a normal distribution and this distribution can be computed with minimal effort, then the outlier can be defined as k times the standard deviation from the mean, where k serves as a threshold. Further investigation into this issue will be conducted in future research. 

## 5. Conclusions

In this study, we propose FR-PatchCore, a feature-level registration method that leverages PatchCore, one of the most advanced anomaly detection methods on the MVTec AD benchmark. Feature-level registration is employed as a pre-text task, and the features stored in memory are optimized through registration loss to approach an optimal feature representation. Additionally, we introduce an innovative ground-truth segmentation method to address false positives resulting from artificial threshold determination. The performance of FR-PatchCore excels in anomaly detection on MVTec datasets and achieves state-of-the-art functionality on MPDD datasets (datasets exhibiting spatial location differences). Furthermore, our method is validated using the high-speed rail dataset. The experimental results demonstrate that FR-PatchCore effectively handles two types of data, “similar category” and “spatial transformation category”, thereby enhancing the generalization capability of PatchCore. Our approach showcases its potential for industrial anomaly detection applications while maintaining high accuracy for effective generalization purposes. However, during our discussion, we also analyze certain limitations of this approach. When significant depth differences exist between samples within the same category, our approach does not sufficiently structure the registration task to cope with such variations, leaving room for future improvements.

## Figures and Tables

**Figure 1 sensors-24-01368-f001:**
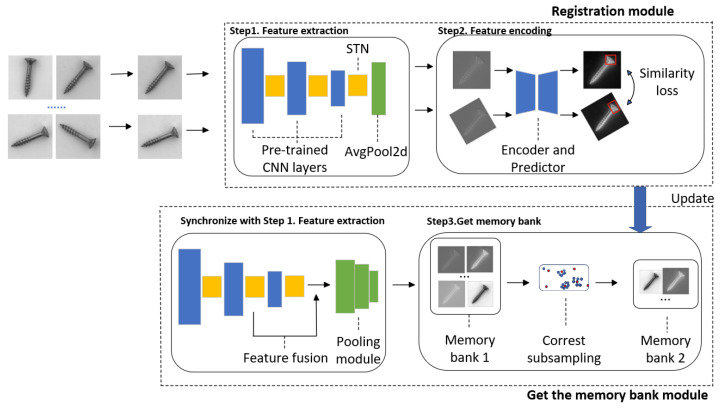
The architecture of the FR-PatchCore model proposed is presented. The entire framework can be divided into three main steps, and the feature extraction processes in both modules are executed simultaneously. In the registration module, randomly selected images are fed into CNN + STN to obtain pre-encoded features. Meanwhile, in the memory bank module, we combine the features passing through the second STN layer and the third STN layer to retain sufficient spatial information of features. Finally, the fusion features are stored in the memory bank.

**Figure 2 sensors-24-01368-f002:**
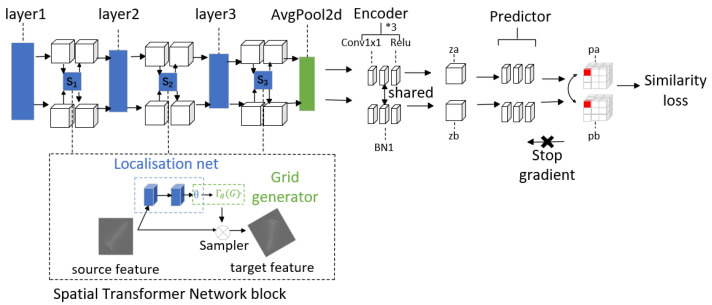
Registration module: two different images of the same category are inputted, and the CNN + STN combination is utilized to extract features. A Siamese structure network is employed for feature encoding and prediction. The objective is to train the registration network in such a way that the predicted features closely resemble the encoded features in the figure.

**Figure 3 sensors-24-01368-f003:**
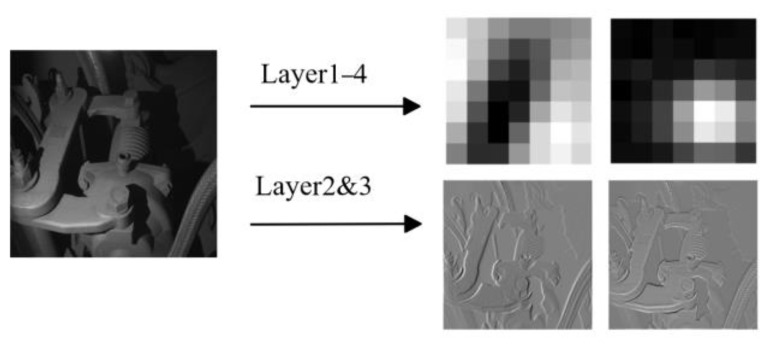
A visualization of features of different levels when fused. When too many levels of the feature are fused, the naked eye can no longer distinguish the outline of the original image, and the features become fine-grained.

**Figure 4 sensors-24-01368-f004:**
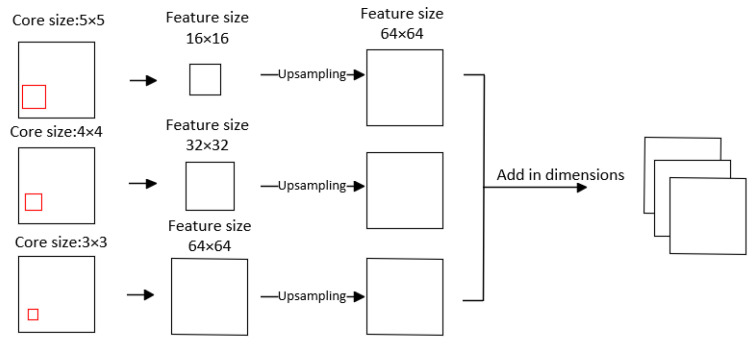
The structure of SPPM.

**Figure 5 sensors-24-01368-f005:**
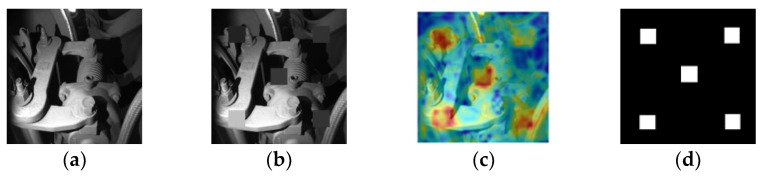
(**a**) source image; (**b**) artificial construction; (**c**) heat map; (**d**) ground truth of the constructed image.

**Figure 6 sensors-24-01368-f006:**
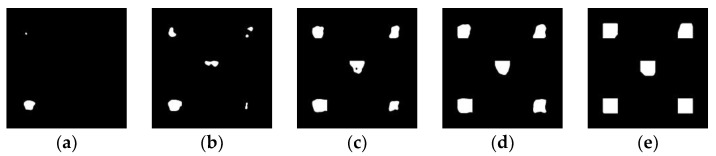
The corresponding anomaly mask under a certain ***th***. (**a**) ***th*** = 9; (**b**) ***th*** = 8.3; (**c**) ***th*** = 7.6; (**d**) ***th*** = 6.4; (**e**) ***th*** = 6.1.

**Figure 7 sensors-24-01368-f007:**
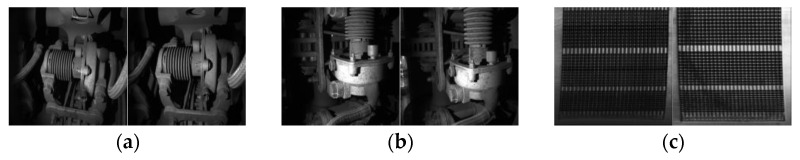
(**a**) HSR dataset 1; (**b**) HSR dataset 2; (**c**) HSR dataset 3.

**Figure 8 sensors-24-01368-f008:**
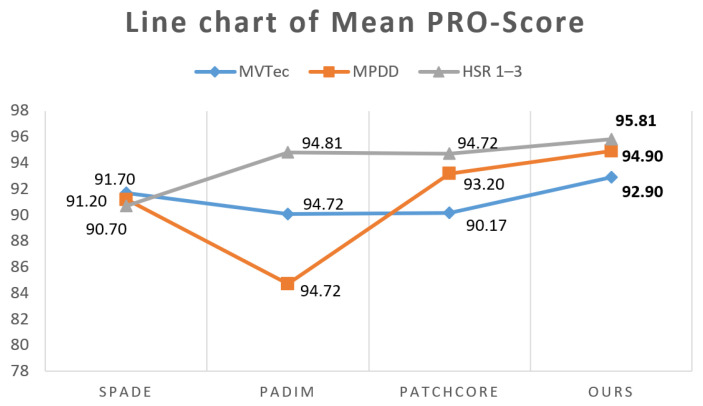
Average PRO score line chart.

**Figure 9 sensors-24-01368-f009:**

Feature visualization extracted by wide_resnet_50_2 network.

**Figure 10 sensors-24-01368-f010:**
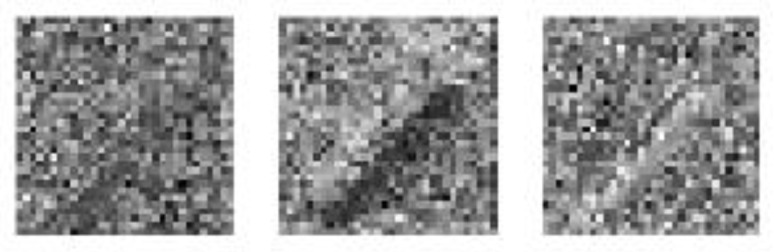
Feature visualization extracted by VIT network.

**Figure 11 sensors-24-01368-f011:**
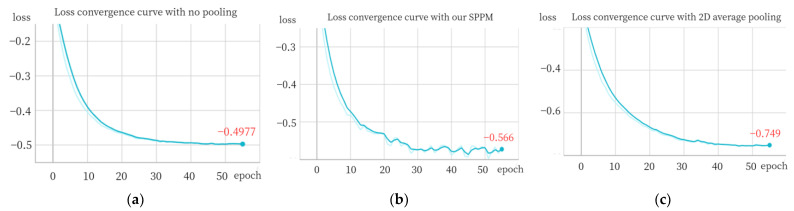
(**a**) Loss convergence curve with no pooling; (**b**) our SPPM; (**c**) 2D average pooling.

**Figure 12 sensors-24-01368-f012:**
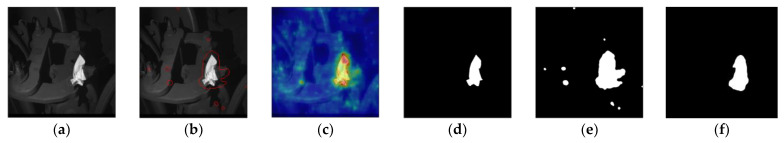
(**a**) Original image; (**b**) segmentation result, the red part represents the initial segmentation of abnormal areas; (**c**) predicted result; (**d**) ground truth; (**e**) the mask has been segmented without the use of any post-processing methods; (**f**) the mask has been segmented using a post-processing method.

**Figure 13 sensors-24-01368-f013:**
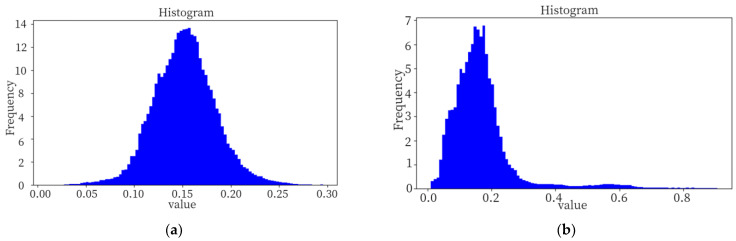
(**a**) Responsiveness histogram (normal); (**b**) responsiveness histogram (abnormal).

**Figure 14 sensors-24-01368-f014:**
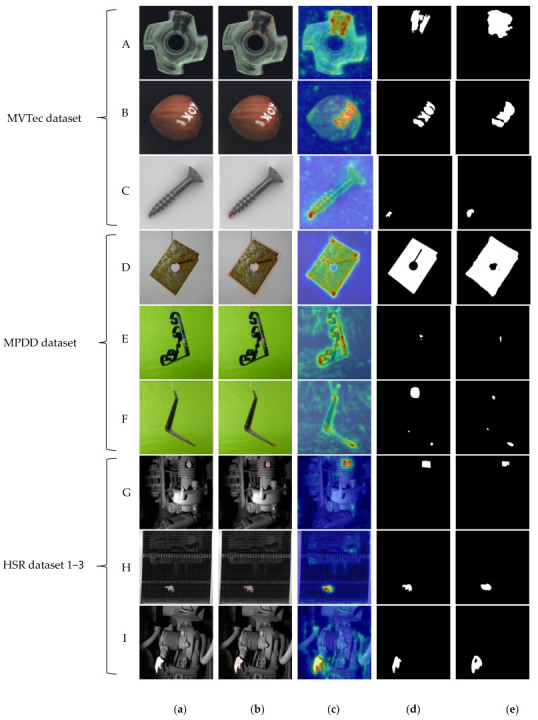
(**a**) original image; (**b**) segmentation result, the red part represents the initial segmentation of abnormal areas; (**c**) predicted result; (**d**) ground truth; (**e**) segmented mask.

**Figure 15 sensors-24-01368-f015:**
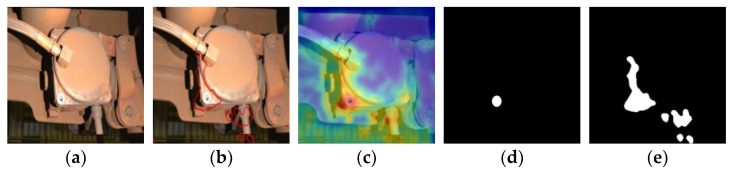
(**a**) original image; (**b**) segmentation result, the red part represents the initial segmentation of abnormal areas; (**c**) predicted result; (**d**) ground truth; (**e**) segmented mask.

**Figure 16 sensors-24-01368-f016:**
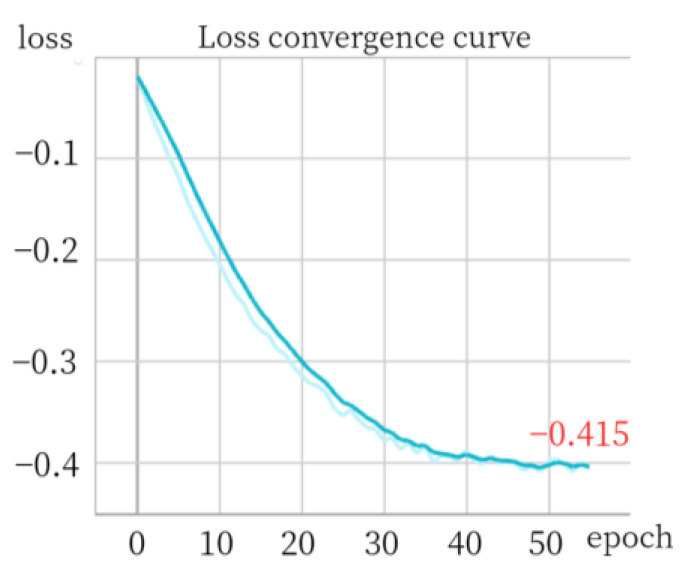
Registration loss function curve of Figure 15a.

**Table 1 sensors-24-01368-t001:** The results on the MVTec dataset (AUROC—image%, AUROC—pixel%).

Category	SPADE		PaDiM		SVDD		Cutpaste		RegAD		PatchCore		Ours	
	Det.	Seg.	Det.	Seg.	Det.	Seg.	Det.	Seg.	Det.	Seg.	Det.	Seg.	Det.	Seg.
carpet	92.80	97.50	**99.40**	**98.90**	92.90	92.60	93.10	98.30	98.50	**98.90**	98.00	**98.90**	97.13	98.35
grid	47.30	93.70	95.70	94.90	94.60	96.20	**99.90**	**97.50**	91.50	88.70	97.50	**97.50**	98.65	95.14
leather	95.40	97.60	99.80	99.10	90.90	97.40	**100.0**	**99.50**	**100.0**	98.90	**100.0**	99.10	99.84	98.84
tile	96.50	87.40	97.40	91.20	97.80	91.40	93.40	90.50	97.40	95.20	**99.40**	94.90	98.46	**97.67**
wood	95.80	88.50	98.80	93.60	96.50	90.80	98.60	**95.50**	**99.40**	94.60	98.90	93.60	98.73	95.45
bottle	97.20	**98.40**	99.80	98.10	98.60	98.10	98.30	97.60	99.80	97.50	**100.0**	98.30	99.76	98.39
cable	84.80	97.20	92.20	95.80	90.30	96.80	80.60	90.00	80.60	94.90	99.30	98.90	**99.78**	**98.17**
capsule	89.70	**99.00**	91.50	98.30	76.70	95.80	96.20	97.40	76.30	98.20	97.60	98.50	**98.83**	98.82
hazelnut	88.10	**99.10**	93.30	97.70	92.00	97.50	97.30	97.30	96.50	98.50	**100.0**	98.40	99.59	97.96
metal nut	71.00	98.10	95.80	96.70	94.00	98.00	99.30	93.10	98.30	96.90	**99.70**	97.70	99.67	**98.77**
pill	80.10	96.50	94.40	94.70	86.10	95.10	92.40	95.70	80.60	97.80	95.90	97.70	**98.82**	**98.21**
screw	66.70	**98.90**	84.40	97.40	81.30	95.70	86.30	96.70	63.40	97.10	94.90	98.60	**96.86**	98.50
toothbrush	88.90	97.90	97.20	**98.70**	**100.0**	98.10	98.30	98.10	98.50	**98.70**	**100.0**	97.20	98.06	98.63
transistor	90.30	94.10	97.50	97.20	91.50	97.00	95.50	93.00	93.40	96.80	**99.90**	94.90	98.96	**97.26**
zipper	96.60	96.50	90.80	98.20	97.90	95.10	99.40	**99.30**	94.00	97.40	99.20	98.40	**99.58**	97.79
average	85.41	96.03	95.20	96.70	92.07	95.71	95.19	95.97	91.21	96.67	98.69	97.51	**98.81**	**97.86**

**Table 2 sensors-24-01368-t002:** The results on the MPDD dataset (AUROC—image%, AUROC—pixel%).

Category	SPADE		PaDiM		SVDD		Cutpaste		RegAD		PatchCore		Ours	
	Det.	Seg.	Det.	Seg.	Det.	Seg.	Det.	Seg.	Det.	Seg.	Det.	Seg.	Det.	Seg.
bracket black	62.80	**97.70**	75.60	94.20	60.17	89.46	77.50	85.20	67.30	72.58	68.15	90.55	**85.98**	96.74
bracket brown	88.00	**98.50**	85.40	95.20	64.56	89.69	**93.40**	93.80	69.60	93.25	74.96	86.46	83.50	95.40
bracket white	83.60	97.00	82.20	98.10	50.89	89.48	74.60	82.70	61.40	95.68	72.56	98.31	**88.25**	**99.50**
connector	90.70	98.60	91.70	97.90	94.52	90.98	97.60	96.80	94.90	92.16	80.95	96.74	**99.95**	**99.11**
metal plate	91.70	97.40	56.30	93.40	92.80	92.70	88.00	92.50	90.20	89.26	89.98	96.30	**98.48**	**98.22**
tubes	51.00	97.50	57.50	92.10	22.15	85.81	**98.20**	97.40	67.90	92.27	69.07	94.11	81.25	**98.85**
average	77.97	97.78	74.78	95.15	64.18	89.69	88.22	91.40	71.90	89.20	75.95	93.75	**89.57**	**97.97**

**Table 3 sensors-24-01368-t003:** The results on the HSR dataset (AUROC—image%, AUROC—pixel%).

Category	SPADE		PaDiM		SVDD		Cutpaste		RegAD(k = 8)		PatchCore		Ours	
	Det.	Seg.	Det.	Seg.	Det.	Seg.	Det.	Seg.	Det.	Seg.	Det.	Seg.	Det.	Seg.
1	98.78	90.70	98.67	93.70	95.46	93.25	78.70	85.27	92.25	90.75	98.28	97.25	**99.32**	**96.58**
2	97.25	95.26	97.68	97.25	98.56	98.25	**99.25**	**98.75**	94.87	95.76	98.26	99.27	98.86	98.86
3	95.45	94.28	96.87	97.36	94.28	96.28	98.36	97.16	98.23	96.14	97.52	97.25	**99.28**	**99.38**
average	97.16	93.41	97.74	96.10	96.10	95.93	92.10	93.73	95.12	94.22	98.02	97.92	**99.15**	**98.27**

**Table 4 sensors-24-01368-t004:** Average PRO scores on three datasets.

Datasets	SPADE	PADIM	PatchCore	Ours
MVTec	91.70	90.10	90.17	**92.90**
MPDD	91.20	84.70	93.20	**94.90**
HSR 1–3	90.70	94.81	94.72	**95.81**
Mean score	91.20	89.87	92.69	**94.54**

**Table 5 sensors-24-01368-t005:** Industrial inspection score on HSR dataset 1.

Metrics	SPADE	PaDiM	PatchCore	Ours
Recall%	88.00	96.00	94.00	**96.00**
Accuracy%	87.80	97.60	**97.92**	97.39
FAR%	12.00	4.00	**3.52**	4.00
TIMES(s)	0.37	0.35	**0.20**	0.22

**Table 6 sensors-24-01368-t006:** Industrial inspection score on HSR dataset 2.

Metrics	SPADE	PaDiM	PatchCore	Ours
Recall%	84.00	94.00	96.00	**98.00**
Accuracy%	82.60	93.04	92.17	**93.91**
FAR%	16.00	6.00	4.00	**2.00**
TIMES(s)	0.42	0.34	0.33	**0.30**

**Table 7 sensors-24-01368-t007:** Industrial inspection score on HSR dataset 3.

Metrics	SPADE	PaDiM	PatchCore	Ours
Recall%	86.00	92.00	96.41	**98.00**
Accuracy%	86.09	92.17	97.92	**98.45**
FAR%	14.00	0.08	4.00	**2.00**
TIMES(s)	0.31	0.26	**0.21**	**0.21**

**Table 8 sensors-24-01368-t008:** Five train iterations on HSR dataset 1.

Iteration	SPADE	PADIM	PatchCore	Ours
1	83.33	88.89	94.44	94.44
2	94.44	94.44	88.89	100.00
3	83.33	83.33	94.44	88.89
4	100.00	94.44	100.00	94.44
5	88.89	100.00	94.44	94.44
Mean Recall	89.99	92.22	94.44	94.44

**Table 9 sensors-24-01368-t009:** Five cross-validation iterations on HSR datasets 1–3.

Iteration	SPADE	PADIM	PatchCore	Ours
1	94.78	98.26	95.65	98.26
2	95.65	93.04	96.52	95.65
3	92.17	93.91	94.78	93.91
4	97.39	95.65	94.78	97.39
5	90.43	94.78	99.13	96.52
Mean Accuracy	94.08	95.13	96.17	**96.35**

**Table 10 sensors-24-01368-t010:** Mean error calculation.

Iteration	SPADE	PADIM	PatchCore	Ours
Mean error	5.92	4.87	3.83	**3.65**

**Table 11 sensors-24-01368-t011:** Comparison of different pooling (AUROC—image%, AUROC—pixel%).

Metrics	No Pooling		SPPM		AvgPool2d	
	Det.	Seg.	Det.	Seg.	Det.	Seg.
scores	95.50	88.50	96.80	90.60	**98.20**	**91.70**

**Table 12 sensors-24-01368-t012:** Module ablation (AUROC%).

Module	MVTec		MPDD	
	Det.	Seg.	Det.	Seg.
No SPPM	95.50	97.51	96.80	90.60
no Registration	95.50	97.64	96.80	90.60
neither	95.50	97.51	95.50	88.50
Both	**95.50**	**97.92**	**95.50**	**88.50**

**Table 13 sensors-24-01368-t013:** The mask generated by the non-post-processing method and the mask generated by our method.

Datasets	$\bar{IoU}$ (Before)	$\bar{IoU}$
HSR dataset 1	0.42	0.68
HSR dataset 2	0.33	0.75
HSR dataset 3	0.37	0.71

## Data Availability

Data underlying the result presented in this paper are available in Refs. [13,14]. Other data are not publicly available at this time but can be obtained from the corresponding author upon reasonable request.

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
