# Peer review of "FR-PatchCore: An Industrial Anomaly Detection Method for Improving Generalization"

_sensors, 2024, doi:10.3390/s24051368_

Round 1
Reviewer 1 Report
Comments and Suggestions for Authors
The article firstly introduces the alignment of feature levels as a self-supervised pre-training task, which enables the model to learn the similarity and spatial location relationship between samples, thus improving the generalization ability and detection effect of the model. Finally, a new method for calculating mask thresholds is proposed to enable the model to scientifically determine the optimal thresholds and accurately classify anomaly masks. Before possible publication, the following main issues are desired to be addressed:
1、For the article proposed wide_resnet_50 and 215 STN of the first three layers of the combination to extract features, but for the article feature extraction process described in the article is not detailed, there is no specific instructions on how to extract, please combined with the experimental process to restate.
2、The article uses Euclidean distance as a calculation method for anomaly scores, but Euclidean distance measures the absolute distance between two points in a multidimensional space and does not take into account the correlation between the data features, is there any other method that would be better to use as as a calculation method for anomaly scores? Is that how you think about this issue? Please give the discussion and analysis by referring to the paper, e.g., doi:10.1109/TGRS. 2022.3217329.
3、The framework diagrams need to be more standardized, they are too simple and scribbled, they need to be more specific and more aesthetic.
4、The article does not give the running time of the experiment and the hardware configuration used, as well as an in-depth analysis and interpretation of the results. It is recommended that the authors add a description of the running time of the experiments and the hardware configuration used in the experimental section to illustrate the efficiency and realizability of the methodology, as well as an in-depth analysis and explanation of the experimental results to illustrate the effectiveness and robustness of the methodology. Please refer to the paper for discussion and analysis, for example, doi:10.1109/JSTARS.2023.3327346
5、The article does not provide a discussion of the limitations and shortcomings of the methodology. It is suggested that the authors add a complexity analysis of the method in the conclusion section to illustrate the efficiency and scalability of the method, as well as a discussion of the limitations and shortcomings of the method to illustrate the scope of application and the direction of improvement.
6、There are still some issues with the language and expression of the article, which need to be revised and embellished. The authors are advised to scrutinize and revise the language and expression of the article in order to improve its readability and professionalism.
Comments on the Quality of English LanguageCan be improved
Author Response
Dear reviewer,
please check the attachment and wish you all the best.

Reviewer 2 Report
Comments and Suggestions for Authors
1. The authors used the wide resnet as the backbone model for feature extraction, but the wide resnet was proposed in 2016. Why did the author choose this way? There is a lack of experiments to illustrate the impact of feature extraction models on detection results.
2. The reason why the registration module chooses negative cosine similarity loss is not stated.
3. In the anomaly detection results (Table 1), the results of the proposed method are not better than the PatchCore method in many categories. What is the reason for this phenomenon.
Comments on the Quality of English LanguageThe English language quality of this paper is high.
Author Response
Dear reviewer,
Please see the attachment, wish you all the best.

Reviewer 3 Report
Comments and Suggestions for Authors
The authors have introduced “FR-PatchCore: An Industrial Anomaly Detection Method for Improving Generalization.” Before considering it for publication, it is imperative to thoroughly examine and address specific limitations and potential drawbacks. Some of the concerns are outlined below:
1. In real-world industrial scenarios, where samples of the same class vary in positions, PatchCore struggles with positional relationships, hindering its ability to handle variations like rotation, flipping, or misaligned pixels. To address this, additional details on proposed strategies or enhancements are needed for a clearer method adaptation to varying sample positions.
2. FR-PatchCore is presented as a solution to the localization issue, the passage does not explicitly discuss potential challenges or limitations that may arise with the proposed method in different contexts or datasets. It would be essential to conduct further evaluations to determine the generalizability of FR-PatchCore across various datasets and industrial scenarios.
3. The study needs to enhance clarity by providing information on model performance parameters, incorporating cross-validation results, and detailing the time complexity of the model. Include metrics that quantify the success and efficiency, such as task completion time, accuracy, or other relevant task-specific measures.
4. It is crucial to explicitly specify the data ratio used for both training and testing phases in order to provide transparency and facilitate a clear understanding of the experimental setup. Additionally, to strengthen the paper’s robustness, a more comprehensive discussion on the generalizability of the proposed approach across diverse datasets and applications is needed. Providing detailed cross-validation results, including results for each cross-validation iteration, would significantly enhance precision and readability
5. Include the full form for any abbreviations used in paper for improved comprehension.
6. In Figure 6, it would be beneficial to explicitly state the threshold value employed for clarity in understanding the presented results. Additionally, there seems to be ambiguity in the depiction of the generated mask from the input. The visual representation appears unclear and misaligned. It is recommended to revise and update this representation to accurately reflect the mask generation process, ensuring a more coherent and visually understandable presentation.
7. To ensure a more accurate representation of results, it is recommended to employ the Mean Intersection over Union (MIoU) metric for evaluating test images.
8. In Table 1 as well as in other Tables, to improved readability and clarity, consider including AUROC-Image% and AUROC-Pixel% under each category. The current presentation might be confusing for readers, and providing these additional metrics directly beneath each category will offer a more organized and comprehensible format.
9. It is advisable to include the equations corresponding to each performance metric utilized in the study. Providing the equations will not only enhance transparency but also offer readers a comprehensive understanding of the methodologies behind the performance evaluation.
10. In all Figures, it appears to be confusing, particularly in terms of the alignment between input, segmented output, and the presented mask. The visual representation does not seem to accurately match the original alignment. To address this concern, it is crucial to revise and accurately depict the relationships between input, segmented output, and the associated mask. Consider highlighting these relationships distinctly to underscore the enhancements made by the proposed model.
Comments on the Quality of English LanguageFor improved readability and clarity, the paper's language structure needs comprehensive editing.
Author Response
Dear reviewer,
Please see the attachment and wish you all the best.

Round 2
Reviewer 3 Report
Comments and Suggestions for Authors
The authors have made an effort to revise the paper, aiming to enhance clarity and readability. However, certain challenges still require more precise attention. The following points outline some of these concerns:
1. The revised representation in Figure 6 highlights the necessity for adjusting the threshold in the proposed model to optimize anomaly detection for each image. Given the dynamic nature of anomaly sizes, it is recommended that the authors integrate an adaptive threshold mechanism to enhance the model's effectiveness in real-time scenarios. Unfortunately, the current model, as outlined in Eq. 10, lacks the capability to sufficiently address this challenge. It is imperative that this concern be addressed more comprehensively to render the model more practically applicable in real-time scenarios.
2. There is a discrepancy in the outcome concerning Figure 6 on pages 375 to 376; a thorough review and correction are necessary.
3. In Figure 15, it is recommended to interchange the sub-figure captions for 'e' and 'd.' It would be more effective if the ground truth image is presented first, followed by the segmented mask. Additionally, the results depicted in this figure are less compelling, especially when comparing them to Figure 6(d). The final outcome appears noticeably divergent from the actual defect portrayed in Figure 6(e), suggesting that the proposed methodology may not be as efficient and practical as desired. The authors are encouraged to reconsider and refine their approach for more accurate findings. Moreover, while the paper reports remarkable results in Table 8, 9, 11 among some others, the visual representation falls short of practical application. The discrepancy between the reported numerical outcomes and the visual representation raises concerns about the real-world applicability of the proposed methodology. It is crucial for the authors to bridge this gap and ensure that the visual results align more closely with the impressive numerical findings, thereby enhancing the overall credibility and practical relevance of the research.
4. Some figure in the manuscript exhibits noticeable fuzziness; therefore, it is recommended that the authors replace it with high-resolution images for enhanced clarity. It would be advantageous if vector graphics or images with a higher resolution are employed to ensure better visual quality and overall readability. This improvement will contribute to a more polished presentation, facilitating a clearer understanding of the depicted content.
5. Please use the journal template for references in this paper and avoid "et al." if there are few authors.
Comments on the Quality of English LanguageModerate editing of English language required
Author Response
Thank you for your excellent work! Please see the attachment.
